# Events Occurring in the Axotomized Facial Nucleus

**DOI:** 10.3390/cells11132068

**Published:** 2022-06-29

**Authors:** Kazuyuki Nakajima, Takashi Ishijima

**Affiliations:** 1Graduate School of Science and Engineering, Soka University, 1-236 Tangi-machi, Tokyo 192-8577, Japan; e21d5903@soka-u.jp; 2Glycan & Life Systems Integration Center, Soka University, Tokyo 192-8577, Japan

**Keywords:** facial nerve, axotomy, motoneurons, microglia, astrocytes, GABAergic neurons, tissue remodeling

## Abstract

Transection of the rat facial nerve leads to a variety of alterations not only in motoneurons, but also in glial cells and inhibitory neurons in the ipsilateral facial nucleus. In injured motoneurons, the levels of energy metabolism-related molecules are elevated, while those of neurofunction-related molecules are decreased. In tandem with these motoneuron changes, microglia are activated and start to proliferate around injured motoneurons, and astrocytes become activated for a long period without mitosis. Inhibitory GABAergic neurons reduce the levels of neurofunction-related molecules. These facts indicate that injured motoneurons somehow closely interact with glial cells and inhibitory neurons. At the same time, these events allow us to predict the occurrence of tissue remodeling in the axotomized facial nucleus. This review summarizes the events occurring in the axotomized facial nucleus and the cellular and molecular mechanisms associated with each event.

## 1. Introduction

The central nervous system is constructed from heterogenous cell types (i.e., neurons and glial cells), and these cell types are considered to interact closely with each other. These cellular interactions are likely mediated by many molecules acting as information molecules or signaling molecules under normal and pathological conditions. Peripheral nerve injury models, which often involve lesions from the transection of facial nerves [1,2] or hypoglossal nerves [3,4], have been shown to appropriately model the cellular interactions between neurons and glial cells. Nonetheless, the details and underlying functions of their interactions have not been adequately clarified.

The facial nerve is the seventh cranial nerve and governs the contraction of muscles for facial expressions [5]. The facial motoneurons exist in the facial nucleus and they extend their nerves to the target muscles across the skull. Facial motoneuron activity is essentially dependent on the commands of upper motoneurons in the cerebral cortex [6,7], and is modified by inhibitory interneurons and presumably glial cells satelliting around motoneuron cell bodies in the facial nucleus [8,9]. 

To accurately analyze the intercellular interaction between neurons and glial cells in vivo, it is desirable to exclude the effects of blood-derived cells and constituents. A rat facial nerve transection model fulfills this purpose, leaving the blood—brain barrier (BBB) unimpaired [1,2,10]. In this model, facial nerve axotomy does not lead to the infiltration of hematogenous cells into the ipsilateral nucleus. This animal model has made it possible to investigate the reactions between neurons and glial cells in the parenchyma without involving the effects of blood-derived cells and blood constituents. This model has also been used to evaluate the effects of survival factors on injured motoneurons [11,12] and to study the regeneration of injured motor nerves [13]. 

In the case of mouse, unlike the rat model, it is known that T cells infiltrate into parenchyma in the axotomized facial nucleus [14,15].

## 2. Damage to Motoneurons

### 2.1. Chromatolysis

When the facial nerve is transected, axotomized motoneurons are retrogradely affected and undergo various changes including morphological changes during chromatolysis [16]. The cell bodies of the injured motoneurons become swollen, and the appearances of the nucleus and cell surface are also changed. Electron microscopic observation has shown that chromatolysis involves the fission of rough-surfaced endoplasmic reticulum and the disaggregation of polyribosomes, disrupting the protein synthesis infrastructure [17]. Quantitative image analysis has revealed that chromatolysis started as early as 8 h post-insult and lasted for 112 days [18]. Thus, chromatolysis is suggested to exert long-term effects on the activity of protein synthesis in injured motoneurons as well as to influence energy metabolism and the expression levels of function-related molecules in injured motoneurons.

### 2.2. Metabolism

After axotomy, the transected motoneurons suffer from damage, but they make efforts to survive and repair themselves. To support this acute emergency, the injured motoneurons appear to increase their energy metabolism. A study using the combination of ^14^C-2-deoxyglucose-injection followed by autoradiography analysis revealed that the amounts of deoxyglucose taken up in the axotomized facial nucleus were increased in rats over the first 28 days after operation [19]. It has been reported that glucose uptake from blood was enhanced in the transected motonucleus in adult rats [20]. Another study indicated that glucose transporter 4 (GLUT4) and GLUT8 are increased in injured motoneurons in the axotomized facial nucleus [21]. These reports suggest that glucose uptake and metabolism are activated in the injured facial nucleus. In the glycolytic pathway, neuron-specific enolase was shown to be enhanced in injured facial motoneurons, contributing to the elevation of energy supply [22]. The levels of glucose-6-phosphate dehydrogenase and 6-phosphogluconate in the hexose monophosphate shunt (the pentose phosphate pathway) were increased in perikaryons in the axotomized facial nucleus [23]. Ribose, a product in the pentose phosphate shunt, leads to RNA synthesis, and another shunt product, nicotine amide adenine dinucleotide phosphate (NADPH), functions to maintain the redox state and/or lipid synthesis in the cells. With regard to lipid synthesis, stearoyl-CoA desaturase (SCD-1), which is an enzyme for the biosynthesis of unsaturated fatty acid, is induced in motoneurons in the axotomized facial nucleus [24]. Based on the above, we speculate that the injured motoneurons make an effort to vigorously synthesize lipids for energy storage.

It has been reported that the metabolism of glycogen/glucose is controlled in the axotomized facial nucleus, since both an enzyme for the synthesis of glycogen (glycogen synthase) and an enzyme for the degradation of glycogen (glycogen phosphorylase) were detected in motoneurons in the facial nucleus [25]. It has long been believed that an energy-storage form, glycogen, is stocked exclusively in astrocytes in the central nervous system [26]. However, in a study by Sotelo and Palay, glycogen was detected in large neurons but not in astrocytes in the brainstem [27]. In fact, glycogen deposit has been detected in motoneurons in the axotomized facial nucleus [28]. The amounts of glycogen have been shown to transiently increase at 3–7 days post-injury, and this increase was explained by a time course analysis of phosphorylated glycogen synthase/glycogen synthase and glycogen phosphorylase. An upregulation of glycogen levels was also observed in transected chicken motoneurons [29] and in developing motoneurons [30]. Given the depression of functional molecules such as choline acetyltransferase (described below) and the increase in glycogen in injured motoneurons, the accumulated glycogen appeared to play a significant role in the response of the injured motoneurons to the acute injury, and the motoneurons probably used the glycogen to generate large amounts of ATP to synthesize the new proteins/RNA/lipids that are a prerequisite for their survival/regeneration.

### 2.3. Function-Related Molecules

Motoneurons produce the neurotransmitter acetylcholine and pack it into the synaptic vesicles that are transported to synapses. Thus, they express an acetylcholine synthetic enzyme, choline acetyltransferase (ChAT), and a vesicle-type transporter, vesicular acetylcholine transporter (VAChT). The ChAT and VAChT are expressed in motoneurons in the facial nervous system (Figure 1A,B) [31,32,33]. In a rat facial nerve transection model, the levels of these marker proteins were found to be significantly decreased in injured but living motoneurons over time until 2 weeks post-insult [34], suggesting that these proteins were downregulated. Although low levels of ChAT and VAChT persist for 2 weeks post-injury, their levels were restored at 4–5 weeks post-injury (Figure 1C). Similar restorations of ChAT levels have been observed in the axotomized hypoglossal nucleus [35,36].

Facial motoneurons innervate the muscles of facial expression, and this innervation is regulated by excitatory and inhibitory neurons. Motoneurons actually express both excitatory glutamate (Glu) receptors and inhibitory gamma-amino butyric acid (GABA) receptors on their surfaces, and the levels of both receptor types were found to be affected by facial nerve transection, similar to the cases of ChAT and VAChT. Ionotropic glutamate receptors (GluRs) including N-methyl-D-aspartate (NMDA) receptors (NR1 and NR2A/2B) and alpha-amino-3-hydroxy-5-methylisoxazole-4-propionic acid (AMPA) receptors were detected in facial motoneurons [37,38]. On the cell membrane, these receptors respond to any excitatory drive including the commands sent from upper motoneurons in the cortex. When the facial motoneurons are injured, the levels of GluR2/3 and NR1 are downregulated [39]. GluR2/3 and GluR4 levels are also decreased in injured hypoglossal motoneurons [40,41]. Not only ionotropic GluRs but also metabotropic GluR mGluR1a have been shown to be decreased in axotomized motoneurons [42]. Based on these results, it seems likely that injured motoneurons tend to refuse excitatory stimuli by reducing the expression levels of the respective receptors.

What is happening in inhibitory receptors in the axotomized facial nucleus? Motoneurons can accept the inhibitory neurotransmitter GABA on specific receptors on the plasma membrane [43,44], leading to hyperpolarization, which blocks the formation of action potential [45,46]. There are two classes of GABA receptors (GABAR). GABA_A_R are ligand-gated ion channels (ionotropic receptors) and GABA_B_R are metabotropic receptors (G protein-coupled receptors) [47,48]. When the rat facial nerve is transected, the levels of the GABA_A_Rα1 subunit are significantly decreased in the axotomized facial nucleus from 3 days to 5 weeks post-injury (Figure 1D) [49]. The levels of GABA_A_Rβ2,3 have been shown to transiently decrease during the period from 5 to 14 days post-insult, but they recovered to the control level at 3–5 weeks post-insult (Figure 1D) [50]. The profile of GABA_A_Rβ1 over a time course resembled that of GABA_A_Rα1 (Figure 1D). A previous study indicated that the mRNA levels of the GABA_A_R subunits (α1, β2, and γ2) and the protein levels of the α1/γ2 subunits were temporarily downregulated in axotomized facial motoneurons [51]. In this study, there was a difference in the recovery time of each subunit; GABA_A_Rα1 was recovered at 60 days post-insult, whereas GABA_A_Rβ2 and GABA_A_Rγ2 were recovered at 30 days post-insult. Similarly, our results indicated that GABA_A_Rβ2,3 was recovered more quickly than the α1 and β1 subunits (Figure 1D). These results suggest that the individual subunits of GABA_A_R in injured motoneurons are not regulated as a group; rather, they are separately regulated by an unknown mechanism.

Acetylcholine receptors are divided into two groups, nicotinic (ionotropic) and muscarinic (metabotropic), based on their pharmacological properties. A metabotropic acetylcholine receptor, m2 muscarinic acetylcholine receptor (m2MAchR), is extensively expressed in the central nervous system including the facial motor nucleus [52]. In the spinal cord, this receptor is known to respond to cholinergic neuron-derived acetylcholine and increases the excitability of motoneurons [53], suggesting that m2MAchR in facial motoneurons serves in the modulation of the excitability of motoneurons. The levels of m2MAchR are severely decreased in the axotomized facial nucleus after 5 days, and m2MAchR remains at low levels until 5 weeks post-insult. In cholinergic synapses, acetylcholine esterase (AchE) exists near the acetylcholine receptors in the cell membrane and there degrades excess amounts of acetylcholine to avoid the strong transmission of acetylcholine, thereby regulating the excitatory stimulus. The level of AchE was found to have declined in the axotomized facial nucleus [23,54], suggesting that the transmission with acetylcholine was reduced in the ipsilateral nucleus.

### 2.4. Survivability and Death

It is generally thought that the susceptibility of motoneurons to nerve transection is age-dependent [55], and younger motoneurons are more highly vulnerable to insult than matured ones. Transection [56,57] or crushing [58] of the facial nerve in infant animals leads to motoneuronal cell death in the ipsilateral nucleus. In an axotomy model of rat, the percentages of surviving motoneurons in axotomized facial nucleus at 7 days post-insult in rats aged 2 days, 1 week, 2 weeks, 4 weeks, and 8 weeks were approximately 10%, 27%, 75%, 102%, and 93%, respectively (Figure 1E) [59]. Motoneuronal cell death was observed between 1 and 3 days post-insult (Figure 1F). Axotomized facial motoneurons of newborn rats are known to regrow the axons and project to the original targets [60]. The endplate basal lamina was suggested to play an important role in the correct guidance of growing axons [61]. In the case of 8-week-old rats, there was almost no change in the number of surviving motoneurons between the control and axotomized facial nuclei at 5 days, 14 days (Figure 1F), and 35 days post-insult. The form of motoneuron cell death is apoptosis, and the apoptosis can be prevented by inhibitors of the mitochondrial permeability transition pore (mPTP) [62] or the overexpression of B-cell leukemia/lymphoma 2 (Bcl-2) [63]. Caspase-3 activation was demonstrated to be important in neuron cell death [64]. As signaling pathways leading to motoneuron death in neonates, the c-Jun N-terminal kinase 3 (JNK3)-related pathway [65] and the zipper protein kinase (ZPK)/dual leucine zipper (DLK) and mitogen-activated protein kinase kinase (MKK) 4 pathways [66] were identified. 

**Figure 1 cells-11-02068-f001:**
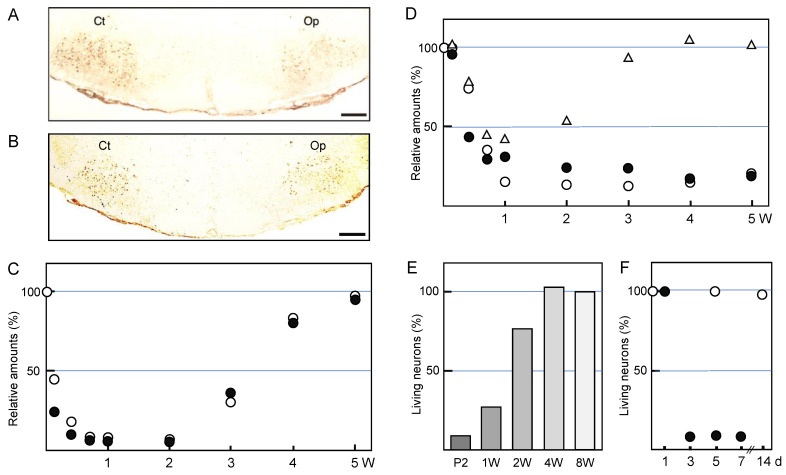
Alterations of injured motoneurons. (**A**,**B**) Immunohistochemistry. The right facial nerve of adult rats was axotomized and the left nerve was left as the control. At five days post-insult, brainstem sections were stained by the anti-choline acetyltransferase (ChAT) antibody (**A**) or by the anti-vesicular acetylcholine transporter (VAChT) antibody (**B**). Ct: control nucleus; Op: axotomized nucleus. Scale bar = 400 µm. (**C**) The time-course transitions of the ChAT and VAChT levels. The right facial nerve of adult rats was transected and the left nerve was left as the control. The brains were recovered at 1, 3, 5, 7, and 14 days, and 3, 4, and 5 weeks post-insult, and the tissue extracts of the control (Ct) and operated (Op) nuclei were analyzed for ChAT (○) and VAChT (●) by immunoblotting. The levels in the Op nucleus are shown as the values relative to that (defined as 100%) in the Ct nucleus. These data were modified from Ichimiya et al. [34]. (**D**) The transitions of the GABA_A_Rα1, GABA_A_Rβ1, and GABA_A_Rβ2,3 levels. Similar to C, tissue extracts of the control (Ct) and operated (Op) nuclei were analyzed for GABA_A_Rα1 (○), GABA_A_Rβ1 (●), and GABA_A_Rβ2,3 (△) by immunoblotting. The levels in the Op nucleus are shown as the values relative to that (defined as 100%) in the Ct nucleus. The results were taken from Kikuchi et al. [49] and Nakajima [50]. (**E**) The survivability of the injured motoneurons; age dependency. The right facial nerve of P2-, 1W-, 2W-, 4W-, and 8W-old rats was cut and the brains were recovered at 7 days. The coronal brainstem sections of each rat brain were stained with the Nissl staining method, and the numbers of living motoneurons were counted. The survival rate in the axotomized facial nucleus is shown as a value relative to that (defined as 100%) of the control nucleus. The results were taken from Koshimoto et al. [59]. (**F**) The time-course changes in neuronal cell death. The right facial nerve of P2- (●) and 8W- (○) old rats was cut and the brains were recovered at 1, 3, 5, 7, and 14 days post-insult. The coronal brainstem sections of each rat brain were stained with the Nissl staining method, and the numbers of living motoneurons were counted, as shown in (**E**).

Although most motoneurons survived in axotomized adult rat facial nucleus as described above, in the case of adult mice, 20–35% of motoneurons died in the axotomized facial nucleus [67]. Another study reported that 20–50% of motoneurons were reduced in axotomized mouse facial nucleus [68]. In the latter case, TNFa and TNF receptors (TNFR1 and TNFR2) were suggested to be involved in the motoneuron cell loss [68].

As noted above, immature motoneurons are vulnerable to nerve injury, therefore the axotomized facial nerve transection model has been used to assay survival factors for motoneurons. The factors displaying survival-enhancing effects are divided mainly into three groups: the ciliary neurotrophic factor (CNTF)/leukemia inhibitory factor (LIF) family, neurotrophins (NTs), and the transforming growth factor beta (TGFβ) superfamily. CNTF [69] was the first survival factor shown to rescue injured infant motoneurons that would otherwise undergo cell death. Subsequently, LIF was found to be a survival factor for motoneurons [70]. As representative NTs, brain-derived neurotrophic factor (BDNF) [71] and NT 4/5 [72] have been shown to contribute to the survival of motoneurons. A TGFβ superfamily protein, glial cell line-derived neurotrophic factor (GDNF), was reported to rescue injured immature motoneurons [12,73]. In addition, acidic fibroblast growth factor (aFGF) [74] and hepatocyte growth factor (HGF) [75] can also rescue injured facial motoneurons. It has been speculated that these survival factors activate a common and specific signaling cascade, leading to the survival of injured motoneurons downstream of each receptor.

The molecular mechanism by which the survival and the death of motoneurons is distinguished in the axotomized facial nucleus should be analyzed in a future study.

## 3. Response of Microglia

### 3.1. Proliferation

The influence of motoneuron insult is somehow transmitted to glial cells located around the motoneuronal cell bodies, and the glial responses have been well-documented [1,2,76]. A striking feature of the microglia response to motoneuronal injury is that it involves increasing the number of cells (Figure 2A), with a peak in the cell number at around 5–7 days post-insult (Figure 2B). The increase in cell number has thus been observed in the axotomized, crushed, or avulsed facial nucleus [77,78] of neonates [77,79,80], adults, and aged animals [81]. Microglial growth in the axotomized rat facial nucleus has been observed in tissue or explant cultures, and the nucleus-derived microglia have been isolated in vitro [82,83].

The increase in microglial cells in the axotomized rat facial nucleus was ascribed to the mitosis of resident microglia. However, very low levels of the infiltration of bone marrow-derived cells into parenchyma were observed in axotomized rat facial nucleus [84]. Thus, it is most likely that the proliferation of resident microglia and small numbers of infiltrated cells contribute to the increase in microglial cells in the axotomized facial nucleus. In vitro experiments have suggested that the factors involved in the regulating mitosis of microglia are colony stimulating factors (CSFs) including macrophage-CSF (M-CSF), granulocyte, macrophage-CSF (GM-CSF), and multi-CSF (interleukin 3; IL-3) [85]. However, among these factors, M-CSF has emerged as the most likely candidate based on a study using osteopetrotic mice (op/op mice) [86] that cannot produce biologically active M-CSF. The facial nerve axotomy of op/op mice resulted in substantially no increase in microglial cell number in the ipsilateral facial nucleus [87]. The M-CSF protein was detected in axotomized facial nucleus-derived microglia [83], and the level was enhanced in the ipsilateral facial nucleus at the time of microglial proliferation [88]. These results suggest that microglia activated by a stimulus induce M-CSF, and M-CSF in turn induces microglial mitosis in an autocrine fashion. However, some reports have demonstrated that M-CSF is produced in neurons. M-CSF was detected in injured sensory neurons [89] and in astrocytes/neurons in the normal spinal cord [90], leaving open the possibility that multiple cell types are engaged in M-CSF production in vivo. With regard to the receptor for M-CSF in microglia, an ^125^I-M-CSF—binding experiment revealed that the level of the M-CSF receptor was increased in the axotomized facial nucleus [91]. They also demonstrated that the M-CSF receptor was induced in the axotomized facial nucleus at the protein/mRNA levels [92]. Subsequently, a protooncogene, cFms [93,94], which encodes the receptor for M-CSF was found to be upregulated in the injured nucleus [88]. Thus, the M-CSF-cFms system was proven to be essential for microglial proliferation in the axotomized facial nucleus. However, in vitro studies have demonstrated that GM-CSF [95] and granulocyte-CSF (G-CSF) [96] exhibit an ability to induce the proliferation of microglia, suggesting that multiple CSFs serve as microglial mitogens in some instances. Furthermore, tumor necrosis factor alpha (TNFα) and interleukin-1beta (IL-1β) were shown to have mitotic capacity in microglia [97]. These cytokines are thought to exert their mitotic capacities particularly in inflammatory regions in the pathological brain. Similarly, IL-4 [98] and IL-5 [99] may also serve as microglial mitogens in inflamed regions.

The molecular mechanism of microglial mitosis was clarified by an analysis of mitosis-promoting factor (MPF), which comprises cyclin and cyclin-dependent protein kinase (Cdk) and advances the cell cycle, leading to cell proliferation [100]. In a rat facial nerve transection model, cyclin A/Cdk2 and cyclin D/Cdk4 were found to serve as the MPFs in proliferating microglia [101]. Proliferating cell nuclear antigen (PCNA), which acts as a processivity factor for DNA polymerase δ in eukaryotic cells and is used as a marker protein of the S phase of the cell cycle, was also induced in mitotic microglia (Figure 2C) [101]. On the other hand, the Cdk inhibitor (CdkI) p21 that serves in cell division arrest [102] was upregulated in the axotomized facial nucleus at the time of microglial proliferation. The upregulation of p21 suggests that the microglial cell cycle advanced by MPF is impeded to not proceed too quickly. In relation to the ability of microglia to divide, a question has been raised as to how many times microglia can divide, or whether microglial mitosis is infinite. An answer was obtained by a study on microglia from the point of view of telomere length. It was clarified that the telomere lengths of microglia are maintained by telomerase [103], and thereby documented the pivotal significance of microglia to the proliferation at lesion sites. 

There is increasing evidence that signaling molecules or signaling cascades participate in the reactions of microglial proliferation. In the facial nerve transection system, cFms stimulated with M-CSF is rapidly phosphorylated and transduces signals to downstream MAPKs including extracellular signal-regulated kinase (ERK), JNK, and p38MAPK [101]. In vitro analysis using specific inhibitors revealed that JNK was linked to the induction of cyclins/PCNA, and p38MAPK was relevant to the upregulation of cFms in microglia. Unlike in the case of the M-CSF-cFms system, GM-CSF stimulation causes the phosphorylation of ERK1/2, Janus kinase 2 (Jak2) and the signal transduction and activator of transcription (STAT) 5A/5B in microglia [104], and activates the hemopoietic cell kinase (Hck) tyrosine kinase and the phosphatidylinositol 3-kinase (PI3K)/protein kinase B (Akt) signaling cascade [105]. TNFα/IL-1β-dependent microglial proliferation was found to be mediated by hydrogen peroxide that was produced by NADPH oxidase [106], suggesting that reactive oxygen species are associated with a signaling cascade, leading to microglial proliferation. 

### 3.2. Morphology

In the normal adult brain, microglia maintain a ramified form with a small soma and a few long prolongations with short and fine projections, but when the brain is injured or diseased, they retract the processes and increase the volume of the cell body, forming activated microglia or reactive microglia [107,108,109]. Starting with del Rio Hortega, many researchers have reported that microglia change their morphology in response to various pathological events [110]. We can observe the morphological change of microglia by means of a facial nerve axotomy model. In the control facial nucleus, microglia remain ramified with some angular long processes (Figure 2D, Ct). These microglia are thought to assume this cellular state in order to sense changes in the environment [1]. On the other hand, in the axotomized nucleus, we can see activated microglia having enlarged and irregular cell bodies with short processes, if any (Figure 2D, Op). This form is thought to be an active state that is adopted to deal with an abnormal alteration in the tissue environment.

The transformation of microglia from a ramified to an activated state is thought to be mediated by specific molecules. Attempts to identify them in vitro revealed that a diverse set of molecules mediates the transformation of ramified microglia to their activated counterparts. These include plasma proteins, growth factors, amino acids, and extracellular matrix (ECM) proteins. Ramified microglia lost their ramification by the elevation of Ca^2+^ and cAMP, or the inhibition of the Gi-protein and phosphatase activities [111]. In contrast, the transformation from the activated form to a ramified form was reported to be mediated by astrocyte-derived soluble molecule(s) [112], astrocyte-derived TGFβ, M-CSF/GM-CSF [113], and glial substrata such as fibronectin and laminin [114]. Retinoic acid, ATP, vitamin E, Ca^2+^, and adenosine have also been reported to modulate the ramification in microglia. It appears that the morphology of microglia in vivo is maintained by a variety of soluble and insoluble factors, some of which are released from astrocytes. 

### 3.3. Migratory Property

Following axotomy of the facial nerve, the cell number and localization of microglia are largely changed in ipsilateral nuclei, allowing us to suggest that microglia can migrate in the tissue. Microglia actually have the ability to move in the axotomized facial nucleus [115,116]. The process in which activated microglia gather around the perikaryon of injured motoneurons was assumed to be mediated by a specific chemokine and the cognate receptor. Microglia have been shown to express chemokine receptors (CCRs) such as CCR1, CCR2, CCR3, CCR5, CXCR4, CX_3_CR1, and IL-8R [117,118]. Among them, CX_3_CR1 in microglia was highlighted as an important chemokine receptor for interacting with injured motoneurons, because the cognate ligand fractalkine was expressed in injured facial motoneurons [119]. However, CX_3_CR1-knockout experiments led to the conclusion that the fractalkine/CX_3_CR1 system is not a substantial mediator between injured motoneurons and microglia in the axotomized facial nucleus [120]. A relationship between monocyte chemoattractant protein-1 (MCP1) and CCR2 was also presumed in the axotomized facial nucleus [121], since the level of MCP1 is enhanced in injured motoneurons, and MCP1 has been implicated in the cellular interaction with microglia expressing CCR2. Thus, microglial satelliting in the axotomized facial nucleus is not fully explained by the relationship between chemokines and specific receptors, leading to the possibility that another molecular mechanism is provided for the cellular interaction.

### 3.4. Metabolic Features

Activated and proliferating microglia in the axotomized facial nucleus appeared to actively serve as neuroprotective and/or immunoregulatory cells. Such microglia were predicted to consume large amounts of energy to perform their functions. Studies of microglial metabolism have described that microglia use glucose, fatty acid. and glutamine (Gln) as major energy substrates [122]. In microglia, glucose is uptaken by GLUT3 and GLUT5 [123,124]. Microglia express acyl-CoA synthetases that catalyze a β-oxidation of fatty acid into acyl-CoA, and they also express high levels of lipoprotein lipase, thus suggesting that microglia use fatty acid as well as glucose to produce ATP [122]. Gln is known to be used as an energy source in the brain, and microglia highly express glutaminase, glutamate dehydrogenase, and soluble linked carrier family 1 member 5A (SLC1A5), preferably metabolizing Gln to generate ATP [122]. Lactate is used as an energy molecule alongside glucose in the brain; lactate is thus taken into the cells by the function of the monocarboxylate transporter (MCT). The MCT was detected in human microglia [125], suggesting that microglia use lactate as a respiratory substance in the facial nucleus. It is conceivable that microglia in vivo developed the ability to generate ATP for their mitosis and other vital activities by using diverse substrates. 

### 3.5. Neurotrophic Property

Observations of the distribution of activated and proliferating microglia enclosing injured motoneuronal cell bodies in the axotomized facial nucleus have focused attention on the function of the microglia because this cell type appears to be directly or indirectly implicated in the health/vitality of motoneurons [78,126]. Inspired by such studies, other reports have investigated microglia from the viewpoint of effector cells on neurons, and consequently, a variety of neurotrophic molecules have been identified in microglia in vitro [127]. Neurotrophic factors are classified into three groups: NTs, the TGFβ superfamily, and the CNTF/LIF/IL-6 family. NTs identified in microglia include nerve growth factor (NGF), BDNF, NT-3, and NT-4/5 [128,129,130]. TGFβ superfamily members detected in microglia include TGFβ1 [131] and GDNF [132]. CNTF/LIF/IL-6 family proteins detected in microglia were CNTF [133] and LIF [134]. As other microglia-derived neurotrophic factors, HGF [135], insulin-like growth factor-1 (IGF-1) [136], and IL-3 [137] have also been reported. An in vitro investigation into the potential interaction between microglia and neurons demonstrated that microglia enhance the production of NGF, NT4/5, TGFβ1, GDNF, FGF2, and IL-3 by stimulating neurons [138], suggesting that neuronal stimulus can activate microglia to produce larger amounts of factors.

TGFβ1 was found in the activated microglia of the injured facial nucleus [139]. TGFβ1-deficient mice demonstrated that TGFβ1 served in neuroprotection as well as microglial proliferation [140]. Whether factors other than TGFβ1 are upregulated in microglia in the axotomized facial nucleus is not clear. However, the following reports might suggest that microglia contribute to the supply of neurotrophic factors to injured motoneurons in a paracrine fashion in the axotomized facial nucleus. It is reported that the mRNA levels of a BDNF receptor, TrkB, are enhanced in injured motoneurons from 2 days to 3 weeks after axotomy [141]. In this experiment, BDNF mRNA was found to peak at 1–2 days post-injury. The mRNA of GDNF receptor components (GDNFR-alpha and c-ret) was increased at 1–3 days post-insult, but GDNF mRNA was not detectable [142]. LIF-R beta mRNA/protein was upregulated in motoneurons after axotomy [143]. Although the upregulated receptors of TrkB, GDNFR, and LIF-R beta in axotomized facial motoneurons allowed us to predict that injured motoneurons attempt to capture ligands released from surrounding microglia, there is no unambiguous evidence for this prediction in vivo. However, considering that the neurotrophic factors are secretory, it may be hard to find the movement from microglia to motoneurons, even if the microglia produced the factors.

Apart from the ability to produce neurotrophic factors, activated and proliferating microglia have been believed to serve as a neuroprotective cell type. A study reported that the transection of the rubrospinal tract leads to neuronal degeneration in the absence of microglial mitosis and ensheathment of lesioned motoneurons, favoring a view that proliferating microglia serve as neurosupportive cells in vivo [144]. In a cerebral ischemic injury model, selective ablation of proliferating microglia resulted in the exacerbation of ischemic injury with decreasing IGF-1, and conversely, the promotion of microglial proliferation by M-CSF accumulation caused an elevation in neurotrophic IGF-1 in the presence of an increase in microglia [145]. These results strongly suggest that proliferating microglia serve as neuroprotective cells in the central nervous system. Some reports have described an association between microglia and the neuroprotection of T cells. In the axotomized mouse facial nucleus, the invasion of CD4^+^ T cells into parenchyma has been recognized [14,15]. The infiltrated T cells were shown to exert neuroprotection of motoneurons. This neuroprotection is conjectured to be induced by IL-10 receptor-expressed CD4^+^ T cells stimulated by IL-10, which is produced in microglia [15,146]. Another report described the non-neurosurvival effects of microglia. The analysis of M-CSF—deficient mice (op/op mice) indicated that the survival of injured motoneurons was not affected in the axotomized facial nucleus [147], suggesting that microglia are not involved in the process of neurosurvival. However, it might be considered that there is a compensatory system in the facial nucleus in that reactive astrocytes serve as neurosupportive cells when microglia are absent.

### 3.6. Protective Ability against Excitotoxicity and Oxidative Stress

Microglial satelliting around injured motoneurons in the axotomized facial nucleus was analyzed from the standpoint of neuroprotective properties, and the activated microglia were suggested to prevent neuronal cell death caused by excitatory Glu toxicity [148]. If Glu is highly released or maintained for a long time in the excitatory synapse, ion channel-type GluRs in postsynaptic neurons are stimulated to an extraordinary degree, and the neurons undergo cell death [149]. To prevent such neuronal cell death, Glu transporters in the nervous system function to remove the excess Glu in the excitatory synaptic cleft and maintain a suitable Glu concentration. Glial cells express two types of Glu transporters, the glutamate aspartate transporter (GLAST; EAAT-1) and glutamate transporter-1 (GLT-1; EAAT-2), and these are thought to be the main functional Glu transporters for protecting against excitotoxity [150,151]. An immunohistochemical analysis of the axotomized facial nucleus revealed that proliferated and satelliting microglia express high levels of GLT-1 at 3–7 days post-insult [152], suggesting that the microglia remove Glu locally by promoting the GLT-1 level. In vitro experiments demonstrated that microglia can uptake ^14^C-Glu time-dependently and in a cell number-dependent manner, and this uptake is suppressed in the presence of a specific GLT-1 inhibitor, dihydrokinic acid (DHK) [153], verifying that microglia uptake ^14^C-Glu through GLT-1. The activity of microglia in the uptake of Glu was enhanced by co-culture with neurons or by the addition of neuronal conditioned medium (NCM), and the enhancement of Glu uptake with NCM was found to be due to the upregulation of the GLT-1 level in microglia [154], suggesting that a neuron-derived soluble molecule(s) provokes microglia to enhance the elimination of Glu by increasing the GLT-1 level. The ability of neurons to elevate the level of GLT-1 has been observed not only in microglia, but also in astrocytes [155]. Microglia isolated from the axotomized facial nucleus actually exhibit the ability to uptake Glu and metabolize it to Gln by Gln synthetase [156]. The series of results described above suggest that GLT-1-expressing microglia protect injured motoneurons from abnormal Glu-induced excitation.

Generally, oxidative stress is implicated in the pathogenesis of neurological disorders including degenerative diseases [157,158]. In the axotomized facial nucleus, the injured motoneurons undergo increased oxidative stress [159]. On the other hand, it is reported that microglia can protect against oxidative stress through cell–cell interactions [160]. Do microglia play a role in protecting injured motoneurons against oxidative stress? A previous paper reported that microglia have the ability to prevent the oxidative stress of neurons in an in vitro experiment using a neuron–glia co-culture system [161]. Natural resistance-associated macrophage protein 1 (Nramp1) [162] and heme oxygenase 1 (HO1) [163] expressed in microglia are presumed to regulate the uptake of iron, and these molecules serve as anti-oxidant and anti-inflammatory enzymes, contributing to the protection of neurons from neurotoxicity. However, whether the anti-oxidative action of microglia is constitutively operated and confers neuroprotection remains under discussion because microglia can exhibit both oxidative and anti-oxidative properties [164,165]. 

### 3.7. Hazardous Property

In the past, microglia were regarded as malign actors in the central nervous system, based in part on their killing of neurons and oligodendrocytes in vitro. Moreover, the analysis of microglial properties indicated that this cell type produces neurotoxic molecules including superoxide anions [166] and nitric oxide (NO) [167]. Later, proteinases and cytokines were grouped into cytotoxic molecules produced by microglia, in addition to reactive oxygen intermediates and reactive nitrogen intermediates [168]. The excitotoxin Glu, eicosanoids, and vasoactive histamine were also demonstrated to be released from microglia. Regarding Glu-toxicity, in the normal state, microglia effectively incorporate Glu by the Na^+^-dependent Glu transporter (GLT-1), which exchanges intracellular K^+^ and extracellular Na^+^/H^+^ and Glu, thereby uptaking extracellular Glu [169]. However, in pathological conditions including ischemia, infection, and neurodegenerative diseases [170], it is suggested that the extracellular K^+^ concentration is elevated, so the transporter reversely transports Glu, resulting in the increase in Glu concentration in the extracellular space [171]. In this case, microglia are regarded as hazardous cells. These microglia-produced neurotoxic molecules and their respective actions have been described in previous reviews [127,172]. Among the various neurotoxic molecules above-mentioned, the reactive oxygen species (ROS), which include superoxide anion, hydroxy radicals, and hydrogen peroxide, are the most dangerous [173]. It has been well-demonstrated that microglia induce superoxide anions in response to endotoxin lipopolysaccharide (LPS) or phorbol ester in vitro, through the action of NADPH oxidase (Nox) [174]. Microglia can produce large amounts of NO in response to LPS or beta-amyloid through the function of inducible NO synthase (iNOS) [175]. NO reacts with superoxide anion to produce peroxynitrite, which is severely toxic to all cell types. These radicals are believed to inhibit respiratory enzymes, oxidize the SH group of proteins, and facilitate DNA injury, finally leading to neuronal cell death [176]. Although it has been suggested that microglia might play a role in the production of ROS and NO in vivo, there is no clear evidence that these harmful radicals are produced by activated microglia in the axotomized facial nucleus.

### 3.8. Inflammatory Property

Microglia have been viewed as immunoregulator or immunomodulator cells in the nervous system [177,178,179], and thus they display a capacity to induce several cytokines [180,181,182]. Among the microglia-induced cytokines, there has been a particular focus on inflammatory cytokines such as TNFα, IL-1β, and IL-6, since they are closely associated with neuronal cell death or inflammation in degenerative diseases including Alzheimer’s disease [183], Parkinson’s disease [184], amyotrophic lateral sclerosis [185], multiple sclerosis [186], and acquired immunodeficiency syndrome dementia [183]. These inflammatory cytokine mRNAs are induced in the axotomized facial nucleus [187]. The findings suggest that motoneuron injury stimulates microglia/astrocytes to induce inflammatory cytokines as mediators of the interaction between neurons and glial cells [188]. This conjectured function of inflammatory cytokines led us to investigate which glial cell type is responsible for the induction of each cytokine and what signaling pathway is related to the induction of each cytokine. 

In vitro experiments have revealed that microglia are the major producers of TNFα, IL-1β, and IL-6 in rats (Figure 2E), although there may be limits to the extensibility of this finding to other animal models, because the particular cell type producing each cytokine differs among species. The analysis of signaling molecules clarified that TNFα, IL-1β, and IL-6 are induced by the JNK/p38MAPK, ERK/JNK, and ERK/JNK/p38MAPK pathways, respectively [189]. IL-1β/IL-6 were found to be induced by a function of NO, while TNFα was induced by superoxide anions, and not by NO [190]. Protein kinase C alpha (PKCα) was involved in the induction of all three of the inflammatory cytokines TNFα, IL-1β, and IL-6. Nuclear factor kappa B (NFkB) was related to the induction of IL-1β in the microglia. Thus, each inflammatory cytokine was found to be induced through a different and a specific signaling pathway. At the same time, other studies have shown that reactive oxygen and reactive nitrogen species induce specific cytokines [189,190]. In this context, the key concept is that the radicals serve in the intracellular signaling cascade, as previously described [191]. 

It is generally accepted that the inflammatory cytokines TNFα, IL-1β, and IL-6 are induced primarily in activated microglia, and vigorously upregulated in the severely affected brain regions [192]. However, in the environment of the axotomized facial nucleus, only limited amounts of TNFα and IL-1β appear to be produced [187]. Thus, TNFα and IL-1β, together with M-CSF, may switch on the proliferation of microglia at an early time-point post-insult, rather than functioning in the inflammatory reaction, since TNFα and IL-1β can potentiate microglial mitosis [97]. On the other hand, a pleiotropic cytokine induced in the axotomized facial nucleus, IL-6 [139], is believed to contribute to the activation of astrocytes and the proliferation of microglia [193]. 

### 3.9. Phagocytic Ability

Phagocytosing cells have been observed in various pathologically damaged sites of the central nervous system such as those affected by Alzheimer’s disease, Parkinson’s disease, multiple sclerosis, ischemia, infectious diseases, toxin-injected brains, and trauma. These cells have been presumed to derive from resident microglia in vivo [194,195,196,197]. However, whether or not microglia change into phagocytic cells depends on the state of the neurons in their vicinity. If microglia sense dead or dying neurons in their proximity, they will be activated and transform to phagocytes, but they will not do so if the neurons are alive or merely injured. A typical example of microglia transforming into phagocytes can be seen in the axotomized facial nucleus in neonates [79,80]. This is because juvenile motoneurons readily undergo cell death in the axotomized facial nucleus [59,60]. In P2-old rats, axotomy-derived motoneuronal cell death occurs between 1 and 3 days post-insult (Figure 1F). Similar to the facial nerve transection model, motoneuronal cell death was observed in the transected newborn rat hypoglossal nucleus [198] and in the newborn rat sciatic nerve transection paradigm [199]. In response to the motoneuronal cell death, the resident microglia are transformed into phagocytes to remove the wreckage of dead cells. This event, performed by the phagocytes, is essential for the early step of tissue remodeling following neuronal cell death. The phagocytic feature of microglia can be detected by the expression of cluster of differentiation 68 (CD68), which is a lysosomal protein [200]. The expression of CD68 in microglia tells us that the microglia serve as active phagocytes with abundant lysosomes. In fact, the transection of the newborn rat facial nerve has been shown to lead to the vigorous induction of anti-CD68 antibody-positive cells in the ipsilateral nucleus (Figure 2F) [79], suggesting that a high level of motoneuronal cell death occurred. In the axotomized facial nucleus, the anti-CD68 antibody-positive cells were Iba1-expressing microglia (Figure 2F), and a fixed population of microglia were anti-PCNA antibody-positive, inferring that the phagocytic microglia were mitotic. In contrast to neonates, adult motoneurons are resistant to the insult, and the vast majority of injured motoneurons do not die if their nerves are transected [59,60]. However, this may not be the case for the intracranial transection and the administration of toxic substances. Unlike extracranial resection, intracranial transection led to a massive neuronal cell death in the ipsilateral nucleus [201]. The injection of Ricinus communis-derived toxic ricin [10] or cholera toxin B-saporin [202] into the facial nerve caused motoneuronal cell death. In the case of ricin injection, the resident microglia were transformed into phagocytes in response to neuronal cell death. Similarly, the administration of cycloheximide into the cut facial nerve resulted in the emergence of phagocytes in the transected facial nucleus [203]. These results indicated that resident microglia have the ability to change into phagocytes, but this capacity is usually concealed in the normal state of the central nervous system. 

How do microglia recognize dead or dying neurons in the facial nucleus? It is assumed that activated microglia can sense neuron-derived “find me” and “eat me” signals [204,205,206], which are released from or induced in dead or dying neurons. At the start of neuronal cell death, the dying cells express an “eat me” signal on the cell surface and release a “find me” signal. Attracted by the “find me” signal, the activated microglia would approach dying cells, and notice the “eat me” signal on the cell surface. Phosphatidylserine-bound chemokine has been identified as one of the “find me” signals [206]. A key “eat me” signal is phophatidylserine itself [207], although intercellular adhesion molecule-3 and carbohydrates also serve as “eat me” signals [205]. As molecules recognizing “eat me” signals, growth arrest-specific gene 6 (Gas6), protein S (ProS), milk-fat globule epidermal growth factor 8 (MFG-E8), and type I membrane proteins have all been reported [205]. As a molecule mediator between dying neurons and microglia, damage-associated molecular pattern (DAMP) has been reported [208]. DAMP proteins are released from dying neurons and stimulate microglia to change into a pro-inflammatory type through the activation of signaling molecules including Toll-like receptor 4, NADPH oxidase, and NFkB [208]. Through this mechanism, activated microglia move closer to the dying cells and engulf them by using various kinds of receptors. Classically, microglia have been thought to eat dead cells by using mannose receptors, complement receptors, and Fc receptors [209]. However, it has come to be understood that the mechanism of phagocytosis comprises many complicated reactions [210]. Thus, microglia are considered to phagocytose dead cells by using certain types of receptors including scavenger receptors and members of the T-cell immunoglobulin mucin domain (TIM) protein family, in addition to the classical receptors. The ATP receptors P2X6 [211] and P2X7 [212] have also been shown to function as scavenging receptors in microglia. Collectively, these studies suggest that, in neonates, the activated microglia that arise in the axotomized facial nucleus can sense “find me” signals released from dying motoneurons, and then can sense “eat me” signals on the motoneurons, and can phagocytose the motoneurons via certain types of receptors. 

It will be interesting to identify the molecule or molecules that are emitted from injured motoneurons and that induce microglial proliferation.

## 4. Response of Astrocytes

### 4.1. Activation

Both astrocytes and microglia respond to facial motoneuron insult and undergo morphological change in the axotomized facial nucleus. The resident astrocytes become hypertrophic, and change their appearance from a protoplasmic to a fibrous form [213]. Therefore, the retrogradely affected astrocytes are called “activated” or “reactive astrocytes. In a normal state, the astrocytes express low levels of glial fibrillary acidic protein (GFAP), which is an intermediate protein expressed specifically in astrocytes [214], but when the facial motoneurons are transected, resident astrocytes are activated and upregulate the levels of GFAP (Figure 3A). The synthesis of GFAP in the axotomized facial nucleus is enhanced as early as 24 h after the axotomy [215], and the increased levels can be seen for 5 days to 8 weeks (Figure 3B). Indeed, another study observed that the upregulation lasted up to 1 year post-insult [216], indicating that the astrocytic activation is maintained for a long period of time [217]. Since astrocytes, unlike microglia, do not proliferate in the axotomized adult facial nucleus [79,218], the amount of GFAP per activated astrocyte is increased, and this activity of concentrating GFAP is an essential role of reactive astrocytes in consolidating the cell structure. The analysis of GFAP-knockout mice demonstrated that GFAP is necessary in order to physically support the cellular construction of astrocytes [219]. A basic role of GFAP in the hypertrophy of activated astrocytes was uncovered through an analysis of GFAP/vimentin-deficient mice [220]. In addition, along with the upregulation of GFAP, another report showed that connexin 43 was also increased in activated astrocytes in the axotomized facial nucleus [221], supporting the notion that the activated astrocytes propagate certain information to neighboring astrocytes by gap junctions. Ultrastructural analysis indicated that the reactive astrocytes bore lamellar processes that enclosed the motoneuron cell bodies for a long period post-insult [222]. The long-lasting astrocytic reaction is speculated to help regenerating motoneurons avoid a shortage of energy materials and/or neurotrophic factors. Aside from this, there is a possibility that the reactive astrocytes disturb the functional regeneration of motoneurons via the obstruction of synaptic connections around the motoneuron cell bodies by astrocyte-derived lamellar processes.

### 4.2. Neurosupportive Features

Activated astrocytes in the axotomized facial nucleus have generally been predicted to serve as neurotrophic cells for injured motoneurons by producing neurotrophic factors including CNTF and FGF [223], GDNF [224], and NTs [225]. IGF-1 was actually identified in astrocytes in the axotomized facial nucleus [226]. The pleiotropic factors TGFβ1, TGFβ2, and TGFβ3 are produced by astrocytes [227,228], and these TGFbs are known to exhibit neurotrophic effects [229,230]. However, it is unclear whether the astrocyte-derived TGFβs act on motoneurons as neurotrophic factors in the axotomized facial nucleus. Thus, it would be of interest to determine whether astrocyte-derived neurotrophic factors function effectively in motoneurons. An analysis of transgenic mice that exhibited overexpression of GDNF specifically in the astrocytes revealed that astrocyte-produced GDNF prevented motoneuron cell death in the axotomized facial nucleus [231], clarifying that the neurotrophic factors produced in astrocytes are actually delivered to the injured motoneurons.

From the viewpoint of Glu toxicity, astrocytes are generally regarded as Glu scavengers. Activated astrocytes at the lesion site show high-level expression of components of an active Glu-uptake system—namely, the Glu transporters GLAST and GLT-1 [232,233]. Glu taken up into the astrocytes through the Glu transporters is converted to nontoxic Gln by Gln synthetase [234]. In the axotomized facial nucleus, astrocyte-like cells have been shown to express GLAST mRNA [235], suggesting that astrocytes protect motoneurons by acting as a glutamate scavenger. However, in pathological conditions such as ischemia and virus infection, astrocytes have been suggested to depress the function of the Glu-scavenger by reducing the Glu transporters [236]. In these dangerous situations, microglia were shown to express the Glu transporter as though they compensate for the role of astrocytes [236].

From the viewpoint of oxidative stress, astrocytes are believed to protect neurons by sustaining a redox homeostasis in the environment of the nervous system [237,238]. In vitro experiments showed that astrocytes as well as microglia have an ability to protect neurons from the cell death induced by oxidative stress [161]. This fundamental feature of astrocytes may, to some extent, contribute to the survival of injured motoneurons in the axotomized facial nucleus.

### 4.3. Energy Metabolism

We will next consider the question of how energy metabolism in the facial nucleus is affected by motoneuron injury and to what extent astrocytes play a role in this relation. In general, glucose metabolism is considered to be responsible for maintaining the energy homeostasis in the nervous system: this includes the glucose metabolism by glycolysis, the pentose phosphate pathway, or glycogen turnover [239]. Astrocytes are the central player for the uptake of blood glucose, glycolysis, and lactate production. Traumatic brain injury alters such glucose uptake activity [240,241], in addition to changing the levels of glucose transporters [242] and lactate metabolism [243]. However, few studies have addressed the subject of energy metabolism in facial nerve axotomy. A single study showed that injured motoneurons upregulated the levels of GLUT4 and GLUT8 [21], but there is no information with regard to the role of activated astrocytes in the axotomized facial nucleus. Regarding glycogen metabolism, it is reported that glycogen granules are deposited in injured motoneurons, but not in astrocytes [28]. Although reactive astrocytes have been predicted to actively metabolize glucose/glycogen in the axotomized facial nucleus, there is no evidence to support the vigorous activity of astrocytes. It appears that glycogen granules are produced in motoneurons, but not in astrocytes in the brainstem [27,28]. Summarizing the features of activated astrocytes in the axotomized facial nucleus, it can be said that this cell type serves in neuroprotection by releasing neurotrophic factors, eliminating glutamate toxicity and oxidative stress, and presumably, by regulating energy metabolism. 

The contribution of astrocytes to the regeneration of injured motoneurons remains to be determined.

## 5. Response of Inhibitory Neurons

Axotomy of the facial nerve caused a decrease in Glu receptors and GABA receptors in the injured motoneurons, as noted above. Inhibitory neurons [8,9] were found to respond to the axotomy of motoneurons. GABAergic neurons are major inhibitory neurons that produce GABA by a specific enzyme, glutamate decarboxylase (GAD) [244], and they pack GABA into the vesicles by means of a vesicular GABA transporter (VGAT) [245,246]. When an action potential arrives at an axon terminal in GABAergic neurons, the vesicles containing GABA are fused to the pre-synaptic membrane and GABA is released into the synaptic cleft. The extra GABA is recovered by GABA transporter-1 (GAT-1) in the pre-synaptic membrane. GAT-1 is a transporter for the uptake of GABA. The GABAergic premotor neurons are actually expressed in the facial nucleus [247] as well as the hypoglossal nucleus [248]. The analysis of GABAergic neuron-specific molecules revealed that the levels of GAD, VGAT, and GAT-1 were decreased transiently in the transected facial nucleus from 5 to 14 days post-insult, but they returned to the control levels at 5 weeks post-insult (Figure 3C) [49]. Immunohistochemical study indicated that the GAD protein was expressed in cells smaller than motoneurons, and the staining intensity of these cells was significantly reduced in the axotomized nucleus (Figure 3D, Op) [49]. VGAT and GAT-1 staining were also decreased in the ipsilateral nucleus. These results indicated that GABAergic neurons transiently downregulated the function of the GAD/VGAT/GAT-1-expressing cells in the axotomized facial nucleus. As inhibitory neurons, there are glycinergic (Glycinergic) neurons in addition to GABAergic neurons. Glycinergic neurons are present in the facial nucleus [247], and the receptor for Gly was found to be decreased in axotomized facial motoneurons [39]. Vesicular inhibitory amino acid transporter (VIAAT)/GABA transporter (VGAT) has the ability to pack both GABA and Gly into the vesicles. This VIAAT/VGAT was also indicated to be reduced in the axotomized facial nucleus [49]. These reports suggest that both GABAergic and glycinergic systems are transiently depressed in the axotomized facial nucleus. However, a study reported that the axotomy of rat facial nerve led to the induction of GABA-inducing depolarization and Ca^2+^ oscillations in motoneurons under the downregulation of the K^+^-Cl^−^ cotransporter (KCC2), suggesting that GABA acts on axotomized motoneurons [249]. Why did the axotomized motoneurons respond to GABA? Because injured motoneurons are substantially active at 3 days post-insult. The brain slices for the electrophysiological experiment were prepared at 3 days post-transection, at which time the levels of GABA_A_Ra1 and GABA_A_Rb2,3 were maintained at approximately 75% and 78%, respectively [49,50]. The levels of GAD, VGAT, and GAT-1 in the axotomized facial nucleus at 3 days post-insult were maintained at approximately 78%, 76%, and 72% compared to those in the control nucleus [49]. These results suggested that at 3 days post-insult, the GABAergic system substantially functions in injured motoneurons. 

As described above, GABAergic neurons are transiently downregulated around injured motoneurons. The relationship between the downregulation of the GABAergic system and synaptic stripping remains to be elucidated.

## 6. Tissue Remodeling in the Axotomized Facial Nucleus 

As shown above, a variety of tissue and cellular changes occur in the axotomized facial nucleus, allowing us to speculate that a certain extracellular protease/protease inhibitor system participates in the tissue reorganization. In fact, the findings where the ECM proteins thrombospondin [250] and tenascin-R [251] as well as the protease/protease inhibitors matrix metallopeptidase 12 (MMP12) [252], serine protease inhibitor 3 [253], and urokinase-type plasminogen activator (uPA) [254] were induced in the axotomized facial nucleus suggest that regulated proteolysis takes place in the extracellular spaces. MMP12, which serves in ECM degradation and/or regulation of the extracellular signaling network, was found to be upregulated in injured motoneurons in the axotomized facial nucleus [252]. mRNA of the serine protease inhibitor 3 (SPI-3; also known as serpin-3) was increased in the axotomized facial nucleus as well as in the hypoglossal nucleus. The predicted targets of SPI-3 are serine proteases, presumably serving in the regulation of proteolysis in the extracellular space [253]. Plasminogen activator (PA) activity was transiently detected in the axotomized facial nucleus, consistent with microglial proliferation. The PA was identified as urokinase-type PA (uPA), but not tissue-type PA (tPA) (Figure 3E) [254]. The uPA production in microglia was facilitated by the stimulation of neurons in vitro [255]. A substrate of uPA, plasminogen (PLGn), was found to be produced in microglia [256], suggesting that active protease plasmin is generated by the activation of PLGn with uPA in microglia. PLGn in microglia was quickly released in response to stimulation with ATP [257]. The microglia-derived PLGn/uPA system was speculated to serve in cordoning off the extracellular spaces that are required for the change in the cellular morphology, cell proliferation, and cellular migration occurring in the axotomized facial nucleus. PLGn promoted the neurite outgrowth of explant brain culture [258] and the maturation of mesencephalic dopaminergic neurons [259], suggesting that the extracellular PLGn/uPA system is involved in the regulation of neurite extension and neuronal development. Furthermore, the plasmin-generating system turned out to contribute to the activation of precursors or inactive forms such as pro-uPA [260], pro-collagenase [261], pro-IL-8 [262], and latent TGFβ [263]. These reactions are essential for biochemical processes that are required for remodeling of the facial nucleus. 

The plasmin-generating system is generally thought to be regulated by intrinsic plasminogen activator inhibitor (PAI), which belongs to the serpin family. In the axotomized facial nucleus, PAI-1 was shown to be enhanced in astrocytes [264]. Astrocyte-producible PAI-1 was found to be upregulated by the PLGn released by microglia [265]. Elevation of PAI-1 was also observed in response to the activation of p38MAPK and JNK in astrocytes [266]. Since PLGn and uPA are produced in the microglia and PAI-1 is produced in the astrocytes [264], plasmin-generating activity could be regulated through the cellular interaction between microglia and astrocytes in the axotomized facial nucleus. Furthermore, analysis of the actions of microglia-derived PLGn on astrocytes revealed that PLGn can enhance the production of TGFβ3, but not TGFβ1 or TGFb2 [267]. The induction of TGFβ3 in astrocytes by PLGn was mediated by proteinase activated receptor-1 (PAR-1). Downstream of PAR-1, the phosphatidylinositol-3 kinase (PI3K)-Akt (protein kinase B) pathway was linked to the production of TGFβ3. TGFβ3 is a pleiotropic cytokine that exhibits neurotrophic activity [229,268]. Given these features, activated astrocytes are expected to serve as neurotrophic cells for motoneurons by providing TGFb3 during tissue remodeling. This protease/protease inhibitor system seems to be involved in the “synaptic stripping” observed in the axotomized facial nucleus.

The concept of synaptic stripping can be observed in both the axotomized facial nucleus [269] and the axotomized hypoglossal nucleus [270]. This phenomenon, in which microglia squeeze into the synaptic cleft comprising motoneurons and inputting neurons, is found in various animals [271] including humans [272]. According to a study, glial cells play a leading role in synaptic stripping by expressing major histocompatibility complex (MHC) class I, complement family members, Toll-like receptors, and ECM proteins [273]. The cellular interaction between neurons and microglia during synaptic stripping has been thought to involve the mediation of many different molecules such as growth factors, chemokines, cytokines, and their receptors [274]. However, it remains uncertain whether synaptic withdrawal from the motoneuron cell body is caused by activated microglia or is a neuron-autonomous event [275]. Although this question remains unanswered, the cellular changes seen in synaptic stripping are clearly a part of the process of tissue remodeling. In a study examining the effect of trigeminal input to axotomized facial motoneurons, trigeminal stimulation was found to improve the reinnervation and recovery of axotomized facial motoneurons [276,277], suggesting that trigeminal neurons are connected to injured motoneurons in the axotomized facial nucleus. 

Synaptic stripping is considered to be a complex event involving certain types of cells and the biologically active substances they produce, indicating that more time will be needed to clarify the precise molecular mechanism underlying synaptic stripping. 

## 7. Mediators Serving between Neurons and around Cells

What signaling molecules are stimulated in the motoneurons by nerve injury? In situ hybridization revealed that ERK and mitogen-activated protein kinase kinase (MEK) mRNA were upregulated in the axotomized facial nucleus [278]. Myristoylated alanine-rich C kinase (MARCK) as a substrate of PKC was elevated in injured facial motoneurons [279]. The stimulus of CNTF led to an early phosphorylation and nuclear translocation of STAT3 in lesioned motoneurons in the axotomized facial nucleus [280]. Overexpression of sonic hedgehog (Shh) in neonatal motoneurons resulted in protection against axotomy-induced motoneuronal cell death [281], suggesting that Shh and its receptor Smoothened in motoneurons are linked to signaling cascades, leading to neuronal survival. Active Ras (GTP-Ras) in motoneurons was found to be depressed in the axotomized facial nucleus [282], demonstrating that active Ras is correlated with the regeneration/functional recovery of motoneurons. These studies suggest that some specific signaling molecules may participate in the survival and regeneration of injured motoneurons. With regard to stimuli arising in a transected stump, it is presumed that an “injury factor” is retrogradely carried from the cut site to the cell body [283]. They argued that this injury factor functioned to induce IL-6 mRNA in neurons, based on an experiment in which a colchicine treatment prevented the expression of IL-6 mRNA in neurons. By means of a similar colchicine-administration experiment, another study demonstrated that the activation of STAT3 in injured motoneurons was attributable to CNTF that was retrogradely transported from the lesion site [280]. Thus, the information regarding signaling molecules and signaling pathways that cause motoneuronal alterations is still fragmentary and insufficient. 

When triggered by motoneuronal lesions, microglia and astrocytes become activated and alter their morphology, mitotic activity, and functional state. Inhibitory GABAergic neurons are also induced to change their functional characteristics in response to motoneuronal lesions. What molecules mediate the interaction between the injured motoneurons and the cellular surround? Injured motoneurons are assumed to output a certain stimulus to the cells surrounding them. One group suggested that altered electrophysiological signaling is among the stimuli emitted from neurons early after injury [284]. Motoneuron-derived calcitonin gene-related peptide (CGRP) was thought to be a probable molecule in the mediation of this interaction [285]. In fact, CGRP was documented to activate microglia and astrocytes [264]. ATP released from injured neurons [286], ATP released from neurons in an activity-dependently manner [287,288], and adenosine [289] are also candidates for neuronal stimuli. Chemokines and their cognate receptors are able to mediate between motoneurons and glial cells. Fractalkine-producible motoneurons can gather CX_3_CR1-expressing microglia [290]. The immune regulator CD200 is expressed in neurons, and the specific receptor CD200R is expressed in microglia/macrophages [291,292], suggesting that neurons have the ability to activate microglia through the association between CD200/CD200R.

In addition, it is possible that injured motoneurons change the levels of ion channels and thereby lead to subtle changes in the ion environment to which microglia respond. In axotomized facial motoneurons, upregulation of voltage-gated Na^+^ alpha III and Ca^2+^-dependent K^+^ channel mRNAs has been reported [293], suggesting that the excitability of motoneurons is changed in the injured state. At this time, the surrounding microglia are considered to sense the change and deal with it, since they express the inward rectifying K^+^ channel [294], inward rectifying K^+^ channel/delayed rectifying K^+^ channel/Na^+^ channel [295], and Na^+^ channel [296]. It is plausible that microglial activation is triggered by unique ion channels, which can capture slight changes in the circumstances of ions in response to neuronal insult. 

A variety of molecules have been proposed as the stimuli released from injured motoneurons or as the mediators between injured motoneurons and surrounding glial cells/inhibitory neurons. In the future, it will be of interest to determine whether the injured motoneuron-derived stimulus for activating glial cells is a humoral molecule(s). The axotomy of facial motoneurons would cause the activation/inactivation of specific signaling molecules responsible for the neuronal and glial changes in the ipsilateral facial nucleus. The signaling pathways related to the neuronal and/or glial responses should be analyzed and clarified.

## 8. Conclusions

Axotomy of the facial nerve leads to alterations in the motoneuronal cell body including chromatolysis, upregulation of glucose-uptake and glucose-transporters, and the downregulation of neurotransmitter-related proteins and receptors (Figure 4). In response to the motoneuronal insult, microglia are activated and transiently proliferate via a mechanism involving M-CSF stimulating its specific receptor cFms, followed by the activation of MAPKs and further potentiation of the induction of cyclins/PCNA (Figure 4). The proliferating microglia are suggested to serve as neurosupportive/neuroprotective cells through the production of neurotrophic factors and the expression of the Glu transporter (Figure 4). In some cases, however, dependent on a pathological or injury state, microglia can exhibit hazardous and/or inflammatory features. Astrocytes are also activated and maintain the activated state for a long period of time (Figure 4). They are considered to contribute to the physical support of injured tissue and to neurorepair by supplying energy-generating materials and neurotrophic factors. Around the injured motoneurons, inhibitory GABAergic neurons are affected and temporally depress the functional molecules (Figure 4). These phenomena are considered to be emergency procedures to preferentially save the injured motoneurons and promote the repair and regeneration processes. A variety of events occurring in the axotomized facial nucleus accompany the cellular/tissue reconstruction. The process of so-called tissue remodeling has been attributed to cell biological reactions including extracellular proteolysis (Figure 4). There is increasing evidence regarding the molecular mechanisms by which events in the axotomized facial nucleus are interpreted by signaling molecules or signaling pathways. Further analyses will be needed to clarify the cellular/molecular mechanisms hidden in the intercellular interactions in the axotomized facial nucleus. 

## Figures and Tables

**Figure 2 cells-11-02068-f002:**
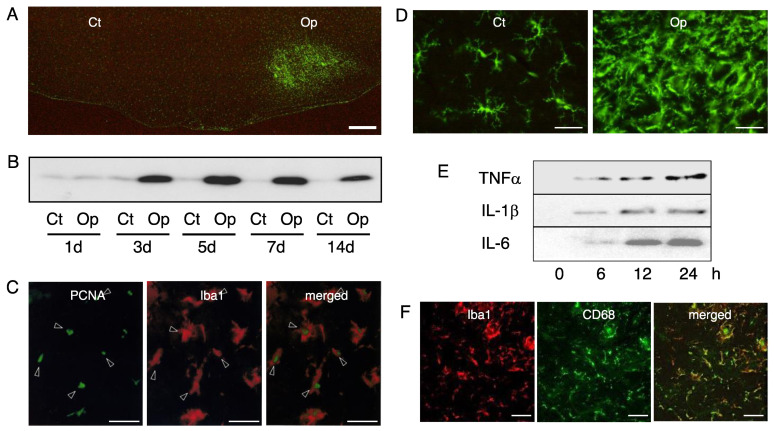
Microglia in the axotomized facial nucleus. (**A**) The response of the microglia to motoneuron injury. The right facial nerve of adult rats was cut and after 5 days, the cryosections of the brainstem were stained by anti-Iba1 antibody. Ct: control nucleus; Op: axotomized nucleus. Scale bar = 400 µm. (**B**) Time-course changes of Iba1 levels. Control nucleus (Ct) and transected nucleus (Op) taken at 1, 3, 5, 7, and 14 days post-insult were analyzed for Iba1 in immunoblotting. (**C**) Mitotic microglia in the axotomized facial nucleus. The axotomized facial nucleus in the brainstem sections prepared at 3 days post-insult was dually stained by anti-PCNA antibody (PCNA) and anti-Iba1 antibody (Iba1). A merged photo is shown at the right. Scale bar = 50 µm. (**D**) The morphology of the microglia in the control and axotomized facial nucleus. Coronal brainstem sections taken at 5 days post-insult were stained by the anti-Iba1 antibody. Microglia in the control nucleus (Ct) and axotomized nucleus (Op) are shown. Scale bar = 50 µm. (**E**) The ability of microglia to induce inflammatory cytokines. Primary microglia (2 × 10^6^/dish) were stimulated with lipopolysaccharide (LPS) and the conditioned media were recovered at 0, 6, 12, and 24 h post-stimulation. Each conditioned medium was analyzed by immunoblotting for TNFα, IL-1β, and IL-6, respectively. (**F**) The transformation of microglia into phagocytes. The right facial nerve of the P2 neonatal rats was axotomized and 3 days later, the brainstem cryosections were prepared. The axotomized facial nucleus was dually stained by anti-Iba1 antibody (Iba1) and anti-CD68 antibody (CD68). A merged photo (merged) is shown at the right. Scale bar = 50 µm.

**Figure 3 cells-11-02068-f003:**
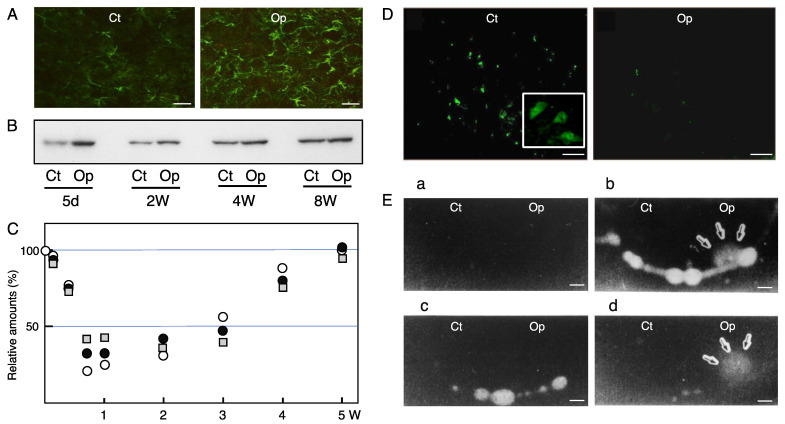
Astrocytes, GABAergic neurons, and uPA in the axotomized facial nucleus. (**A**) The response of astrocytes to motoneuron lesion. The right facial nerve of adult rats was transected and after 6 weeks post-insult brainstem sections were subjected to immunohistochemistry for GFAP. Ct: control nucleus; Op: axotomized nucleus. Scale bar = 50 µm. (**B**) The time-course changes of GFAP The control nucleus (Ct) and transected nucleus (Op) of the brainstem collected at 5 days, 2 weeks, 4 weeks, and 8 weeks post-insult were analyzed for GFAP by immunoblotting. (**C**) The time-course transitions of GAD, VGAT, and GAT-1 levels. The right facial nerve of adult rats was transected and the left nerve was left as the control. The brains were recovered at 1, 3, 5, 7, and 14 days, and 3, 4, and 5 weeks post-insult, and the tissue extracts of the control (Ct) and operated (Op) nuclei were analyzed for GAD (○), VGAT (●), and GAT-1 (□) by immunoblotting. The levels in the Op nucleus are shown as the values relative to that (defined as 100%) in the Ct nucleus. Data were modified from Kikuchi et al. [49]. (**D**) The immunohistochemical staining for glutamate decarboxylase (GAD). Immunohistochemical staining for GAD was carried out in the brainstem sections of rats whose right facial nerve was cut 5 days earlier. Ct: control nucleus; Op: axotomized nucleus. ChAT-expressing motoneurons are shown inside of the control side (Ct) (note that the GAD-expressing cells are much smaller than motoneurons). Scale bar = 50 µm. (**E**) The zymography for uPA in the axotomized facial nucleus brainstem sections of adult rats whose right facial nerve was cut 5 days earlier was analyzed. Four kinds of zymography were performed: zymography in which agarose gel alone (**a**), agarose gel containing PLGn (**b**), agarose gel containing PLGn and plasmin inhibitor amiloride (**c**), or agarose gel containing PLGn and anti-tPA antibody (**d**) was overlaid on each brainstem section. The brainstem sections with each agarose gel were incubated at 37 °C, and at a suitable time point, the agarose gels were stained with amide black. Plasmin activity can be seen as a lytic area. Ct: control nucleus; Op: axotomized nucleus. Scale bar = 500 µm.

**Figure 4 cells-11-02068-f004:**
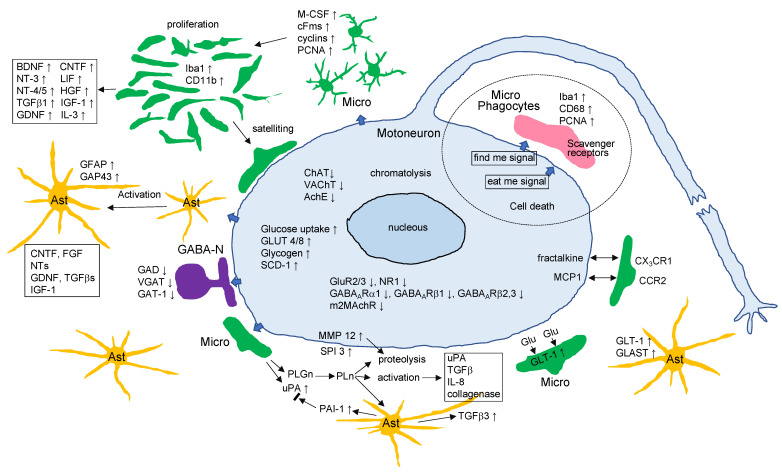
The events occurring in the axotomized facial nucleus. Various events observed in the axotomized facial nucleus are depicted in the diagram. Micro: microglia; Ast: astrocytes; GABA-N: GABAergic neurons. Other abbreviations are as defined in the text.

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
