# Peer review of "Events Occurring in the Axotomized Facial Nucleus"

_cells, 2022, doi:10.3390/cells11132068_

Round 1

Reviewer 1 Report

Manuscript ID: cells-1775304

Type of manuscript: Review

Title: Events occurring in the axotomized rat facial nucleus

Authors: Kazuyuki Nakajima, Takashi Ishijima

The MS is a review of glial cell activation and molecular changes in the facial nucleus (FN) after facial nerve lesions. The authors provide an overview of current knowledge on glial cell reactions and molecular signaling of lesioned neurons to trigger these glial cell changes.

The experimental model based on studying cellular and molecular events in the facial nucleus after facial nerve lesions provided a wealth of important information about the reactions between the axotomized CNS neuron and the surrounding glial cells.

Although the MS is a fairly extensive review of these reactions, the text requires several additions and clarification.

Major objections

It is necessary to comment more on the apparent absence of BBB disruption in the injured FN and the ongoing invasion of CD4+ T cells, which act in concert with microglial cells to support axotomized FN neurons and their regeneration (see, Raivich et al. 1998; Jones et al. 2005). This contradicts the claim by the authors on page 1 that: This animal model has made it possible to investigate the reactions between neurons and glial cells in the parenchyma without involving the effects of blood-derived cells and blood constituents.

Another contradiction is found in the text on page 6, §1: The increase of microglial cells in the axotomized facial nucleus was ascribed to the mitosis of resident microglia rather than the migration from the neighboring place. - in comparison to text of page 7, subchapter 3.3. Migratory property.

What is it really like with an increased number of activated microglial cells in FN after the facial nerve lesion? Is there really no possibility of invading circulating monocytes and transforming them into microglial cells in FN after facial nerve transection? See, Flugel et al. 2001. Therefore, it is necessary to mention and clarify the results of activation of microglia after total transection of the facial nerve and transection of only a branch of the facial nerve.

Pg 11, § 2: How do microglia recognize dead or dying neurons in the facial nucleus? The latest information on the activation of microglial cells by DAMPs released from damaged neurons is also missing from the MS text and should be added. 

Reviewer 2 Report

This manuscript is in the scope of the journal and contains detailed information on the axotomy induced changes in the rat facial nucleus. The summarized data are of potential interest to a broad spectrum of neuroscientists beginning with first-aid neurosurgeons, ENT-physicians and ending with specialists in genetics and biochemistry. It also well written and well illustrated. I have several minor comments and recommendations.

1) Maybe I missed it, but the authors do not emphasize, that the facial nerve axotomy provides a very good model to study many aspects of regeneration in a purely motor nerve.

2) Considering the deafferentation (synaptic stripping of the facial perikarya), the authors could have addressed the trigemino-facial relations, which are to a certain degree also affected by the facial nerve axotomy. Three articles are suitable for basic reading.

Pavlov S, Grosheva M, Streppel M, Guntinas-Lichius O, Irintchev A, Skouras E, Angelova S, Kuerten S, Sinis N, Dunlop SA, Angelov DN (2008) Manually stimulated recovery of motor function after facial nerve injury requires intact sensory input. Exp Neurol, 211: 292-300.

Bendella H, Pavlov SP, Grosheva M, Irintchev A, Skouras E, Angelova S, Merkel D, Sinis N, Kaidoglou K, Dunlop SA, Angelov DN (2011) Non-invasive stimulation of the rat whisker pad improves recovery of whisking function after simultaneous lesion of the facial and trigeminal nerves. Exp Brain Res, 212:65-79.

Skouras E. Pavlov S, Bendella H, Angelov DN (2013) Stimulation of trigeminal afferents improves motor recovery after facial nerve injury. Functional, electrophysiological and morphological proofs. Advances in Anatomy, Embryology and Cell Biology, vol. 213: 1-111, Springer-Verlag, Berlin Heidelberg.

3) Is already clear why 30-40% facial motoneurons in adult mice die after axotomy, but those in adult rats do not ?

4) Is it already clear why the axotomized perikarya of very young (newborn) rats regrow with an astonishing accuracy towards their original targets?

5) I will be very pleased if the authors, apart from their "8. Conclusion" try to offer some advices and recommendations to people dealing with nerve repair at the end of each big chapter.

Reviewer 3 Report

In general, this work is really interesting and well written. My comments aim to increase the scientific soundness and clarity of it.

Introduction – “nervus facialis” instead “the nervus facialis”. But I do not understand the idea of using the Latin name in this case. Does it has any special meaning and introduce something important? Noteworthy, other anatomical terms (like facial nucleus or rubrospinal tract) are written only in English.

2.1. – Please reword the sentence “One of the changes is a morphological change known as chromatolysis [12]” – remove one “change”

2.2 - Please unify a method to referring to a source publication. For example “Ito et al. [16]” could be changed to „It has been also reported [16]”; “Gómez et al. [17]” to “In other studies [17]” etc.

2.3 – Please change “VAchT” into “VAChT”. It is more accurate. By the way, it would be appropriate to abbreviate acetylcholine to “ACh”

Figure 2 and figure 3 – scale bars are missing

General remark – the authors usually use the term "nervous system" which mostly means "central nervous system". But in some sentences it would be beneficial to clarify that the authors mean "central nervous system" but not others, such as "peripheral nervous system" or "enteric nervous system".

Reviewer 4 Report

The review from Nakajima and Ishijima provides a comprehensive update on the retrograde reaction to axotomy of facial motoneurons and the associated glial cells, astrocytes and microglia. The facial nucleus is the most studied model of neuronal injury avoiding issues of direct injury to the CNS or break down of the blood brain barrier. This is an scholarly update that will be certainly useful in the field. However, the review is focused to just one species, the rat, and that leads to some important omissions on mechanisms that might differ between rat and mouse and that I believe they need to be carefully considered. This refers specifically to section 2.4. In here the authors should consider the extensive literature on cell death in adult facial motoneurons in the mouse and that has been used extensively as model to study neuroprotection and neurodegeneration in adult motoneurons. I believe their conclusion of cell death after axotomy is exclusively restricted to the younger ages based in one study from the authors performed in the rat. I also understand they are trying to focus the review to rats, but when there are differences at major as this, I believe it will help the reader if they are mentioned. Moreover, the distinction between both species is blurred because many of the conclusions are based on data from mouse transgenics. Therefore I would also omit rat from the title, since many of the studies are based on mice.

Another issue that usually confounds this reviewer is in section 3.5. Many in vitro studies suggested a large variety and abundance of neurotrophic factors expressed by microglia. However, I am frequently confused on which ones have been validated as expressed and released by microglia activated in situ around injured motoneurons. I would like to see a more detailed discussion on this issue. For example, the authors indicate that motoneurons increase expression of TrkB, GDNF-receptors and Lif-receptors and imply this is an “attempt to capture more ligands released from surrounding microglia”. Is there in vivo evidence that microglia surrounding axotomized motoneurons in situ express the ligands for these receptors?

Other specific issues

Page 3, 2nd paragraph: The authors suggest that AMPA and NMDA receptors are “accepting” commands from superior motoneurons in the cortex. I think the nomenclature used here is non-standard and somewhat cofounding. I guess they intended to say “upper” motoneurons instead than “superior”. On the other hand, these receptors will be responding to any excitatory drive to facial motoneurons and that may include, but is not exclusive of, inputs from cortex.

Page 3, 2nd paragraph: The opening of GABAA receptors in facial motoneuron leads to hyperpolarization or shunting inhibition not the depolarization suggested by the authors, unless, of course if the facial motoneuron is axotomized.

Page 3, 2nd paragraph: the authors discus the GABAA receptor subunits that change expression in facial motoneurons after axotomy, but they failed to report on the subunits that do not change expression. Both are important to consider.

Page 3, 3rd paragraph. In one sentence of this paragraph the authors imply that the facial nucleus is part of the peripheral nervous system. It is not. Is in the CNS, although of course the facial motoneuron axons travels in the PNS. May be this is just a grammar issue, but it is confusing.

Page 3, 3rd paragraph. The authors suggest that the expression of M2 muscarinic receptors on facial motoneurons has unknown function. The authors should look further information in spinal motoneurons were M2 opposite to C-terminals has been shown to modulate the excitability of the motoneuron.

Page 6, source of M-CSF or CSF1. It will be desirable to indicate that in other models of axotomy, like spinal motoneurons and sensory neurons, CSF1- is found exclusively in the injured neurons and not in microglia or other glial cells. The result of the authors in the facial nucleus (with CSF1 expressed solely by microglia) seems unique to the model.

Page 7, 2nd paragraph in section 3.2. The authors suggest that: “Retinoic acid, ATP, vitamin E, calcium and adenosine have also been reported to cause ramification in microglia” I would substitute “case” for “modulate” since these molecules sometimes promote microglia process extension, but in other circumstances they induce retraction, based on context specific interactions that include microglia activation status, phenotype, level of ligand and except receptor or combination of receptors activated.

Page 8, last paragraph in section 3.5. This section discuses microglia neuroprotection mechanism but seem to ignore the extensive works of Dr. K Jones (Indiana) on the role of microglia interaction with T-cells for exerting neuroprotection in the facial nucleus of the adult mouse. Is there any reason why this literature is omitted when discussing microglia neuroprotective mechanisms on facial motoneurons?

Page 9, in two different sections microglia are proposed to remove glutamate to prevent excitotoxicity or to release glutamate and induce excitotoxicity. Can the authors clarify this issue?

Page 13, section 4.2. The neurosupportive action of astrocytes seem to parallel those discussed also for microglia (neurotrophic actions, glutamate scavenging etc…). Is it possible to discuss the relative contribution of each of these glial cells to these mechanisms and whether they are acting at the same times post-injury?

Page 14, Section 5. The authors discuss the downregulation of GABA producing enzymes and vesicular and plasma membrane transporters and postsynaptic receptors suggesting downregulation of GABA neurotransmission, however the electrophysiological evidence is that GABA is very active on axotomized motoneurons. The following work need to be discussed in this paragraph:  Toyoda H, Ohno K, Yamada J, Ikeda M, Okabe A, Sato K, Hashimoto K, Fukuda A. Induction of NMDA and GABAA receptor-mediated Ca2+ oscillations with KCC2 mRNA downregulation in injured facial motoneurons. J Neurophysiol. 2003 Mar;89(3):1353-62. doi: 10.1152/jn.00721.2002. Epub 2002 Nov 13. PMID: 12612004.

Page 14, last paragraph. The authors propose that serpin-3 is upregulated and serves in “synaptic stripping”. However, the cited reference suggest serpin-3 upregulation is specific to the facial nucleus after axotomy and does not occur in the spinal cord, while the “synaptic stripping” phenomena is a general mechanism that occurs in every motoneuron in spinal or brainstem nuclei in all species.

English issues

Page 2: “Quantitative image analysis has revealed that chromatolysis starts much faster (as early as 8 h post-insult) and last for longer (112 days)”. It is unclear what is being compared to. May be the sentence needs rephrasing without comparative adverbs

Page 3, 1st paragraph: Regarding the lower levels of ChaT and VAChT the authors suggest it is being degraded; that will imply actual direct targeting of these proteins. Is this really the word they want to use? How do we know is degraded or just expression downregulated? In other model Chat mRNA is downregulated suggesting lower gene expression and it needs to be kept in mind that Chat and Vacht mRNas are expressed form the same genomic locus

Page 3. When referring to neurotransmitter receptors on the membrane of facial motoneurons I would use the word “respond to” instead than “accept” when indicating what ligands activate these receptors

Page 5 F in figure legend. Neuron is misspelled

In general, the English will benefit from a thorough review
